# Uptake and Adherence to National Guidelines on Postpartum Haemorrhage in Italy: The MOVIE before–after Observational Study

**DOI:** 10.3390/ijerph20075297

**Published:** 2023-03-28

**Authors:** Serena Donati, Marta Buoncristiano, Paola D’Aloja, Alice Maraschini, Edoardo Corsi Decenti, Ilaria Lega

**Affiliations:** 1National Centre for Disease Prevention and Health Promotion, Istituto Superiore di Sanità-Italian National Institute of Health, Viale Regina Elena 299, 00161 Rome, Italy; serena.donati@iss.it (S.D.); marta.buoncristiano@gmail.com (M.B.); paola.daloja@iss.it (P.D.); ilaria.lega@iss.it (I.L.); 2Servizio Tecnico Scientifico di Statistica-Italian National Institute of Health, 00161 Rome, Italy; alice.maraschini@iss.it

**Keywords:** postpartum haemorrhage, guideline adherence, practice guideline

## Abstract

Translating evidence-based guidelines into clinical practice is a complex challenge. This observational study aimed to assess the adherence to the Italian national guidelines on postpartum haemorrhage (PPH) and describe the clinical management of haemorrhagic events in a selection of maternity units (MUs) in six Italian regions, between January 2019 and October 2020. A twofold study design was adopted: (i) a before–after observational study was used to assess the adherence to national clinical and organisational key recommendations on PPH management, and (ii) a cross-sectional study enrolling prospectively 1100 women with PPH ≥ 1000 mL was used to verify the results of the before–after study. The post-test detected an improved adherence to 16/17 key recommendations of the guidelines, with clinical governance and communication with family members emerging as critical areas. Overall, PPH management emerged as appropriate except for three recommended procedures that emphasise different results between the practices adopted and the difference between what is considered acquired and what is actually practised in daily care. The methodology adopted by the MOVIE project and the adopted training materials and tools have proved effective in improving adherence to the recommended procedures for appropriate PPH management and could be adopted in similar care settings in order to move evidence into practice.

## 1. Introduction

Obstetric haemorrhage, despite multiple efforts, is still the leading cause of maternal morbidity and mortality at the global level [1,2,3]. Moreover, bleeding-related events are one of the most avoidable causes of maternal mortality [4]. Although there are several known risk factors for postpartum haemorrhage (PPH), including multiple pregnancies, prolonged third stage of labour, and retained placenta [5,6], many cases of PPH occur unexpectedly [5,6]. Early clinical and/or instrumental diagnosis are fundamental for a prompt treatment. Mechanical measures to stimulate uterine contraction represent the first-line management of PPH [5,6]. Pharmacological (i.e., oxytocic drugs such as oxytocin, ergomentrine, and carboprost, or tranexamic acid) and surgical (e.g., balloon tamponade and compression sutures) measures are effective treatments that can be administered both in sequence or simultaneously [5,6]. The first Italian estimate of the maternal haemorrhagic mortality ratio (MRR) equal to 2.90/100,000 live births between 2000 and 2007 [7] was four times the figure detected in the UK (0.66/100,000 live births) between 2003 and 2005 [8]. Therefore, the Italian Obstetric Surveillance System (ItOSS) of the Istituto Superiore di Sanità (Italian National Institute of Health) implemented a bundle of research and training activities to assist healthcare professionals in the prevention and management of obstetric haemorrhage, with the ultimate goal of reducing avoidable maternal deaths [9]. The bundle comprised four main components: a prospective population-based project on haemorrhagic maternal near misses [10,11,12]; the promotion of multidisciplinary intra-hospital clinical audits in cases of severe haemorrhage; distance learning activities on prevention and treatment of PPH [13]; and the development of the first national evidence-based guidelines on PPH [14]. The bundle approach, expected to increase the uptake and compliance of recommended interventions [15], has proven effective in reducing the haemorrhagic MMR from 2.49 to 0.77 per 100,000 live births in five Italian regions, which have been participating in enhanced maternal mortality surveillance since its inception [9].

The implementation of evidence-based practices remains a critical challenge, and international agencies developing clinical guidelines highlight the need for appropriate strategies to ensure that recommendations are translated into local initiatives for the improvement of clinical practice [16,17]. In light of the positive impact of postpartum haemorrhage management guidelines [18,19] and of quality improvement initiatives in reducing the occurrence and severity of obstetric haemorrhage, ItOSS launched and coordinated, in collaboration with six regions, the project “Monitoring and evaluation of the implementation of evidence-based recommendations for the management of obstetric haemorrhagic emergencies” (MOVIE). The aim of the project was to standardise, monitor, and evaluate the implementation of key clinical and organisational recommendations selected from the national PPH guidelines [14] to improve the quality of care in case of haemorrhagic postpartum events.

The present study aims to evaluate the adherence to the PPH guideline’s key recommendations before and after the implementation of the MOVIE project, and to describe the clinical management of women with PPH ≥ 1000 mL giving birth in a selection of Italian maternity units (MUs).

## 2. Materials and Methods

The project adopted a twofold study design: a before–after observational study to report the adoption of the guideline’s recommendations by the MUs’ chief physicians and contact clinicians, and a cross-sectional study to assess their implementation in daily clinical practice among the participating MUs.

The six regions involved in the project (Piedmont, Emilia-Romagna, Tuscany, Lazio, Campania, and Sicily), covering 49.5% of national births, are distributed in the north, centre, and south of the country and have been participating in the ItOSS surveillance since its inception. The MUs in each participating region were selected as follows: coverage of 20% of regional births; different volume of annual deliveries (<500, 500–999, 1000–1999, ≥2000); different organisational models (hub and spoke facilities); and types of facilities (public, private with agreement, and private).

The study population includes women giving birth in the participating MUs with an estimated blood loss ≥1000 mL, the chief physicians participating in the online survey, and the contact clinicians of the participating MUs notifying PPH incident cases.

The choice of the blood loss cut-off was based on the MUs’ workload feasibility, according to the expected number of cases per MU estimated through the Hospital Discharge Database, taking into account the volume of annual deliveries and the hospital case mix.

### 2.1. Before–after Observational Study

From 1 January to 31 August 2019, an operating unit (OU) was set up in each region to assist the project’s activities. A team of four health professionals (obstetrician, anaesthesiologist, midwife, and risk manager) were enrolled in each of the 43 participating MUs, and a contact clinician was appointed. Clinical and organisational key recommendations (Appendix A) have been selected from the PPH national guidelines [14] in collaboration with a group of national experts. Clinical recommendations explored the following items: early recognition of pregnancies at risk for PPH, adoption of vital parameters monitoring charts (e.g., MEOWS), estimation of blood loss, medical and surgical PPH therapy, and transfusions of blood and blood products. Organisational recommendations explored: organisation of care, protocols for PPH management, clinical risk management, and communication between healthcare professionals and women and family members.

The before–after study adopted two survey tools: an online survey for the MUs chief physicians, and a before–after checklist for the MUs contact clinicians. The online survey adopting a questionnaire designed and reviewed by national experts collected information on the structural and organisational characteristics of the MUs and on the implementation of the selected key recommendations. The before–after checklist reviewed by national experts assessed the implementation of the selected key clinical and organisational recommendations of the national PPH guidelines. The contact clinicians completed the checklists at the start of the project, in May 2020, as a mid-term evaluation, and at the conclusion of the project.

From 1 September 2019 to 31 October 2020, 8 continuing medical education courses were offered to up to 30 participants each. The training courses were dedicated to the MUs multi-professional teams, including the contact clinicians. They were held by ItOSS researchers and by regional project representatives adopting teaching methods for adults (e.g., group work, clinical case-study simulations, and role-play). The following tools were delivered to the participating MUs:A national guideline “Postpartum haemorrhage: how to prevent it, how to cure it” in two versions: for healthcare professionals [14] and for citizens [20];A clinical pathway on the prevention, diagnosis, and management of PPH developed in accordance with the selected recommendations from the national guidelines;A poster describing the flow chart for PPH management reviewed by experienced obstetricians, anaesthesiologists, and midwifes, recognised at the national level;A modified early obstetrics warning system (MEOWS) monitoring and alert chart adapted for the national context;A situation background assessment recommendation (SBAR) methodology [21] proposed as a technique to facilitate communication between professionals in the event of a PPH emergency.

Each MUs contact clinician received all of the teaching materials and the project tools to implement cascade training in their MUs by involving all of the staff.

### 2.2. Cross-Sectional Study

Between 1 November 2019 and 31 October 2020, the contact clinicians notified PPH incident cases by using personal credentials to access the web platform dedicated to data collection. The estimated number of expected cases per MU according to the national Hospital Discharge Database was used to assess the completeness of incident case reporting.

The online form to collect clinical information on the management of PPH was developed by ItOSS and reviewed by national experts. It included the following thematic areas: maternal socio-demographic characteristics; PPH prophylaxis; cause of the bleeding; estimation of blood loss; laboratory investigations; medical and surgical therapy; transfusions of blood and blood products; maternal and perinatal outcomes; and organisational procedures of the MUs. A monthly zero reporting was adopted to distinguish the absence of PPH cases from the failure in reporting.

The following data were analysed: information collected through the online survey compiled by the chief physicians; information collected through the before–after checklists filled in by the MU contact clinicians; and information on incident cases of PPH with blood loss ≥1000 mL inserted in the web platform by the MUs contact clinicians. The frequency distributions of all the categorical variables of interest were calculated. Continuous variables such as age, body mass index (BMI), and blood loss were reduced to classes. The estimate of blood loss was aggregated into three classes according to the severity of the bleeding (between 1000 and 1500 mL; 1501 and 2000 mL; over 2000 mL) and used as a stratification variable to compare case management by PPH severity. Other stratification variables used in the analysis were the cause of the PPH and the geographical distribution of the MUs. A descriptive analysis of the online survey and of the before–after checklists were performed using an Excel spreadsheet.

## 3. Results

All six participating regions joined the initiative. Appendix A describes the annual number of births in the MUs enrolled in each region, according to the 2019 National Hospital Discharge Database, the number of enrolled MUs, and their number per organisational model and type of facility.

Eight training courses were held between September and October 2019. Overall, 167 health professionals (9 chief physicians, 47 obstetricians, 37 anaesthesiologists, 46 midwifes, and 28 risk managers) attended the training, of which 93% acquired the educational credits and rated the quality of the course as 4.6 over 5, where 5 is a ranking of excellent. During the first four months, 95% of the participating MUs reported incident PPH cases. A decrease in data reporting was registered from the beginning of the COVID-19 pandemic in March 2020. Overall, the percentage of cases notified compared with those expected was 93% (*n* = 274) in Piedmont, 70% (*n* = 304) in Emilia-Romagna, 78% (*n* = 293) in Tuscany, 56% (*n* = 229) in Lazio, 19% (*n* = 90) in Campania, and 7% (*n* = 30) in Sicily. The latter two regions were excluded due to coverage <50%. Therefore, data analysis was carried out from a total of 24 MUs located in two regions of northern Italy (Piedmont and Emilia-Romagna) and two regions of central Italy (Tuscany and Lazio).

### 3.1. Online Survey for the Chief Physicians

All chief physicians completed the survey before the project implementation. Overall, the MUs involved in the project had structural and organisational characteristics in line with the reference standards established by the Ministry of Health [22], and no critical situations arose. The multi-professional teams made up of obstetricians, midwives, anaesthesiologists, and paediatricians/neonatologist were on call 24 h a day, as required by the reference standards in all the participating regions. About half of the participating MUs were equipped with a department for pregnancy at risk. All but one of the chief physicians reported the 24 h availability of at least one operating room for obstetrics activities. The rates of caesarean section (CS) performed by the participating MUs reflect that of the geographical area to which they belong, ranging between 10 and 20% in northern Italy and 30 and 40% in central Italy [23].

### 3.2. Before–after Checklists for the MUs Contact Clinicians

In line with the chief physicians’ survey results, the before–after checklists confirmed that many of the investigated care procedures were already in place before the start of the project (Table 1).

Almost all participating MUs (20/24) had shared a clinical pathway for PPH prevention and treatment, 23/24 had shared working instructions with the blood transfusion centre, and all MUs had already in use recommended methods to quantify blood loss. The most critical aspects before the start of the project included: the adoption of a checklist for PPH risk factors (8/24); the availability of shared procedures to deal with women who refuse transfusions (6/24); and conducting clinical audits for cases of PPH > 1500 mL blood loss (7/22). Moreover, before the start of the project, structured procedures to manage communication among health care professionals and families during a PPH emergency (5/24) or during pre-discharge counselling (5/17) were rarely available. At the end of the project, nearly twice the MUs had adopted a communication strategy protocol. Table 1 describes in detail the answers before the start and after the end of the project, highlighting how the intervention resulted in an improvement for all but one recommendation. The mid-term results reported by the contact clinicians in May 2020 are in between the before and after checklist.

### 3.3. Data Analysis of the PPH Cases Notified during the Study Period

From 1 November 2019 to 31 October 2020, the 24 participating MUs notified 1100 cases of PPH ≥ 1000 mL. Uterine atony was the leading cause (n = 667; 60.6%), followed by tissue-related (n = 238; 21.6%) and traumatic causes (n = 129; 11.7%), 4.3% (n = 47) were classified as intra-operative haemorrhage, 1.2% (n = 12) as bleeding due to coagulation abnormalities, and 0.5% (n = 6) due to other causes. Tissue-related bleeding included placental tissue retention (n = 172; 15.6%), abruptio placentae (n = 34; 3.1%), abnormal placentation (n = 26; 2.4%;), and placenta previa (n = 6; 0.5%). The traumatic causes of haemorrhage were genital laceration (n = 105; 9.5%), variceal bleeding (n = 14; 1.3%), uterine rupture (n = 3; 0.3%), and uterine inversion (n = 3; 0.3%).

Most of the reported cases (n = 833; 75.7%) had an estimated blood loss between 1001 and 1500 mL, 17% (n = 186) between 1501 and 2000 mL, and 7% (n = 81) >2000 mL. Table 2 describes the socio-demographic and obstetric characteristics of the cohort. Overall, 39.6% (n = 435) of women underwent CS, recording more often cases of severe PPH (58% in case of loss >2000 mL). Trauma-related PPH was associated with vaginal delivery in 71% (n = 92) of the cases, while coagulation abnormalities resulted in urgent/emergency CS in over 45% (n = 6) of cases. Intraoperatively bleeding complications occurred more frequently in case of elective CS (n = 23; 48.9%) compared to urgent (n = 21; 44.7%) and emergency (n = 3; 6.4%) indications.

### 3.4. PPH Prevention

Overall, 60.3% (n = 663), 61.8% (n = 680), and 55.9% (n = 615) of the women had, respectively, no ante, intra, and postpartum identifiable risk factors for PPH. Among those for which risk factors have been identified, 18% (n = 198) were before the onset of labour, 11.8% (n = 130) intrapartum, and 17.6% (n = 194) postpartum. For 24 women (2.2%), the information was missing. In detail, the most frequent antepartum risk factors were prior hysterotomy, including CS (19.4%), multiple pregnancy (7.0%), and placenta praevia (5.1%). Fast birth (9.2%), episiotomy (8.9%), and CS during labour (8.5%) were the most frequent intrapartum risk factors, while II and III degree perineal laceration (19.3%), withheld placenta (14.1%), and birthweight >4000 g (9.5%) were reported as the most frequent postpartum risk factors. During the postpartum surveillance, monitoring charts for the detection of women’s vital parameters were used overall in 38.8% (n = 417) of the cases (Table 3). A pharmacological PPH prophylaxis with oxytocin was administered in 99.3% (n = 1092) of cases. Methylergometrine and carbetocin were used, respectively, in 4.5% (n = 50) and 1.3% (n = 14) of cases. Regardless of the extent of the blood loss, clinicians used at least one recommended method for estimating the loss in 98.8% of PPH cases (Table 3), mainly transparent graduated bags and gauze and sheets counting. A prepartum haemoglobin (Hb) value was detected, respectively, in 47% (n= 517) of cases during pregnancy and 44.5% at hospital admission; 69% of the women had a fibrinogen dosage available before birth. Among women with one or more ante partum PPH risk factor, 5.5% (n = 24) lacked ante-partum Hb, 92.9% (n = 406) ante-partum platelet dosage, and 77.1% (n = 337) fibrinogen. The availability of the PTT and APTT ratio during bleeding increased with the extent of blood loss (Table 3).

### 3.5. PPH Medical and Surgical Treatment

Medical therapy was administered in 97.2% of cases, oxytocin was used as a first-line drug treatment in 88.1% of cases (n = 969), and ergometrine was administered in 48.4% (n = 532). As second-line treatment, prostaglandins, was administered with a lower prevalence as blood loss increased (Table 3). The administration within three hours of delivery of 1 g of tranexamic acid in association with uterotonics was recorded in 57.8% of the whole sample, more often (82.7%) in the case of severe PPH (Table 3).

In the case of unresponsive PPH to first- and second-line drug treatment, at least one recommended manual procedure was performed in 93.2% of the cases, mostly uterine massage (90.3%; n = 993). Other procedures included uterine curettage in 37.1% (n = 408), bimanual uterine compression in 23.4% (n = 527), balloon tamponade in 15.6% (n = 165), and uterine tamponade in 4.2% (n= 46).

Recourse to surgical procedures and/or embolisation concerned 9.5% (n = 104) of the whole sample, and 39.2% (n = 31) of the women with a PPH > 2000 mL. Haemostatic uterine sutures and embolisation were performed, respectively, in 1.5 (n = 16) and 1.2% (n = 13) of the cases, and hysterectomy in 7.0% (n = 76) of the enrolled women.

### 3.6. Fluid and Transfusion Management

The administration of fluids in the acute phase of the haemorrhage was performed in the majority of women (83.7%; n = 921), regardless of the extent of the loss. Packed red blood cells were transfused in 85.0% (n = 68) of cases with >2000 mL loss compared to 12.1% (n = 100) among those with loss <1000 mL (Table 3). The use of fresh frozen plasma (n = 37) and platelets (n = 9) only concerned the cases with loss >1500 mL.

### 3.7. Maternal and Perinatal Outcomes

Severe maternal morbidity was reported in 1.8% as a consequence of haemorrhagic shock; 8.5% of mothers were affected by severe anaemia. No maternal deaths have been reported. Newborns below 2500 g at birth were 8.9% (n = 98; 12.3% in case of >2000 mL), and four perinatal deaths were reported.

### 3.8. Risk Management

Overall, 89.6% of MUs reported to have an emergency kit for PPH in the delivery room. The anaesthesiologist was called in 90.1% (n = 73) of the more severe cases and in 77.3% (n = 644) of the less severe. During the emergency, family members were informed of the woman’s health state, in percentages, in 21.7% (in case of loss <1500 mL) and 40.7% (in case of >2000 mL loss) of cases. The care pathway was described in the discharge letter of the majority of the cases, ranging between 71.3% in less severe and 91% in more severe cases. Audit activities were reported in 3.5% (n = 29) of cases with loss <1500 mL, in 10.2% (n = 19) of those between 1500 and 2000 mL, and in 27.2% (n = 22) of the most severe cases.

## 4. Discussion

Research has explored potential effective strategies to enhance compliance and assess the implementation of PPH guidelines [17,18,19,24,25]. Quality improvement initiatives including health professionals training, risk assessment protocols, the adoption of flowcharts and checklists, clinical audits, and patient empowerment tools have been implemented with the aim of enhancing clinician’s skills, ameliorated care, and thereby preventing avoidable severe haemorrhagic events [18,25,26,27].

To our knowledge, MOVIE is one of the few proactive programs in Western countries aimed at improving the uptake of and adherence to evidence-based guidelines for PPH prevention and management. In Wales, a set of initiatives based on national guidelines has been introduced to reduce the progression of PPH from moderate to massive [24], resulting in clinically significant improvements across the country. A recent French multicentre study [25] validated the effectiveness of a high-quality protocol for the prevention and management of PPH, including the key recommendations outlined in the national guidelines.

According to the MOVIE baseline survey, structural and organisational characteristics of the participating MUs met the quality and safety standards required by the Italian Ministry of Health [22]. Similarly, the before–after checklist filled in by the MUs contact clinicians highlighted that some care procedures were already widely in place before the start of the project, and that all the investigated procedures recorded an improvement in the post-test. The widespread use of the recommended methods to quantify blood loss, regardless of its severity, highlighted the clinicians’ awareness of the importance of PPH early diagnosis. Clinical audits in the case of PPH and the availability of a checklist for PPH risk factors in the medical records registered a limited improvement. However, experiences of excellence have also been highlighted for the structured monitoring of obstetric near misses at the regional level in the Emilia-Romagna region [28]. Despite the fact that the management of communication between professionals and women and their families betrays the historic lack of dedicated training for clinicians, the MOVIE project allowed for doubling the number of MUs equipped with a structured method for communicating with the woman’s family members during the PPH emergency. There is room for improvement with respect to the recommendations of the guidelines which emphasise the importance of communicating with family members from the onset of the obstetric complication and recommend performing audits for cases of obstetric haemorrhage with a loss of >1500 mL (Table 1).

The cross-sectional study provided detailed descriptions of the clinical pathways adopted for each incident PPH case and the opportunity to verify the real implementation of the guideline’s recommendations in daily clinical practice. Compared to Birth Registry data [23], women of the MOVIE sample were more often >35 years of age (46.2% vs. 35.0%), and instrumental vaginal delivery (6.2% vs. 3.9%), CS (39.6% vs. 31.8%), multiple pregnancy (7% vs. 1.6%), and birthweight > 4000 g (9.5% vs. 5.2%) were more common. The reported high rate of placenta praevia (5.1%) is likely a possible consequence of the high Italian CS rate [23].

The number of severe cases detected by the MOVIE project are too few to allow for a comparison with other studies on severe PPH [24] and with the previous ItOSS project that enrolled haemorrhagic near misses requiring ≥ 4 units of whole blood or packed red blood cells [10]. In fact, during the MOVIE project, total or subtotal hysterectomy was performed in 0.6% and 6.1% of the cases, respectively, while the previous ItOSS study detected a very high rate of peripartum hysterectomy (10.9/10,000 maternities) [10], the highest among nine European countries [29]. The maternal and perinatal outcomes of the MOVIE cohort were overall good, reporting no maternal deaths and 4/1000 perinatal deaths in line with the national rate [23]. According to the cross-sectional study, three recommended procedures resulted in a lower uptake in daily care compared to the before–after checklist results. Graphic charts for monitoring/alerting vital signs were notified only for 37.9% of detected haemorrhages, although the checklist reported its adoption in 58% and 92% of the MUs before the start and at the end of the project. Their use in large-scale clinical practice is not yet widespread in Italy. As recommended by the national guidelines [14], improvement of these tools in terms of standardisation, definition of alarm cut-offs, adaptation to different clinical contexts, and definition of the implementation process is necessary. The uterine Balloon was adopted only in 15% of haemorrhages despite all but one MUs reporting its availability at the start of the project. Tranexamic acid, in addition to the standard uterotonics, was administered, respectively, in 82.7%, 69.9%, and 52.6% of blood loss cases >2000 mL, 1500–2000 mL, and 1000–1499 mL, even though the checklist reported its administration in 54% and 92% of the MUs before the start and at the end of the project, respectively. Given the strong recommendations of the guidelines based on high-quality evidence in support of the early administration of tranexamic acid, the detected difference seems worthy of attention. Both the different timing of data collection among the two sources of data, and the gap between what is considered acquired and what is actually put in place in daily care, could account for the detected differences.

On the contrary, the organisational aspects of care described by the cross-sectional study were in line with the results of the before–after checklist. According to both sources, a shared protocol for maternal surveillance was adopted in over 90% of PPH cases and 90% of the MUs were equipped with a kit for PPH emergencies. The results of the cross-sectional study confirmed the low rate of clinical audits and the need for improved communication with the women and their family members, which are the two main criticalities which were found.

The study also has limitations. The COVID-19 pandemic emergency has compromised the implementation of the planned activities, both at the central and at the local level. ItOSS was unable to monitor the compliance rate, the completeness, and quality of the reports on a regular basis. Among all the participating MUs, those located in Campania and Sicily were unable to continue participation according to the required standards when coordination became less constant. This resulted in exclusion from the study of the MUs in the Italian geographical area with the highest direct maternal mortality ratio [30]. Despite the exceptional nature of the pandemic situation, this might suggest that implementing these kinds of interventions may require dedicated resources in some contexts. A second limitation is the lack of a pre–post evaluation on PPH incident cases or of a control group that prevent us from assessing the impact of the intervention directly on PPH cases.

The MOVIE project, in continuity with the ItOSS bundle of activities supporting the prevention of avoidable maternal mortality and morbidity from haemorrhagic causes [4], promoted multidisciplinary work and highlighted the aspects of care most in need of attention. The project’s strengths include the involvement of the multi-professional teams in a participatory process that facilitated the identification of unmet needs, and the assessment provided by the chief physician’s survey and the before–after checklists and their validation through the cross-sectional study.

## 5. Conclusions

The MOVIE project consolidated the ItOSS network of researchers and clinicians, identified the organisational and clinical areas requiring improvement in the management of PPH, promoted multi-professional collaborations to improve the appropriateness of care, and developed a package of tools to support their implementation, monitoring, and validation. According to the study’s results, the adherence to PPHs key recommendations improved over the study period in most MUs. The adoption of the MOVIE methodology, supported by the developed package of training materials and tools, might contribute to facilitate the uptake of and adherence to evidence-based guidelines on postpartum haemorrhage in similar care settings.

## Figures and Tables

**Table 1 ijerph-20-05297-t001:** Before–after checklists compiled by the contact clinicians at the Maternity Units.

Checklist Item	Before the Start of the Project N = 24		At the End of the Project N = 24	
	Yes n (%)	No n (%)	Yes n (%)	No n (%)
Checklist of risk factors for PPH identification and documentation included in the medical record	8 (33%)	16 (67%)	13 (54%)	11 (46%)
Use of recommended methods for quantifying blood loss (graduated bags, counting of gauze and drapes, etc.)	24 (100%)	0	24 (100%)	0
Use of graphic charts for monitoring/alerting vital signs (e.g., MEOWS)	14 (58%)	10 (42%)	22 (92%)	2 (8%)
Early administration of 1 g of tranexamic acid in addition to the standard therapy with uterotonics in case of PPH	13 (54%)	11 (46%)	22 (92%)	2 (8%)
Availability of intrauterine balloons for the treatment of PPH in the delivery room	23 (96%)	1 (4%)	24 (100%)	0
Periodic training sessions in the use of the intrauterine balloon	12 (50%)	12 (50%)	16 (67%)	8 (33%)
Availability of a procedure with working instructions shared with the blood transfusion centre on how to request and obtain emergency blood, including group 0 blood, Rh D and K negative, in case of PPH	23 (96%)	1 (4%)	24 (100%)	0
Availability of a shared procedure between gynaecologists, anaesthesiologists, obstetricians, and transfusionists for the clinical management of women who refuse transfusion	6 (25%)	18 (75%)	8 (33%)	16 (67%)
Availability of a clinical pathway for the prevention and treatment of PPH	20 (83%)	4 (17%)	23 (96%)	1 (4%)
Availability of a shared procedure between gynaecologists, anaesthesiologists, midwives, and transfusionists for the management of women at increased risk of PPH	19 (79%)	5 (21%)	22 (92%)	2 (8%)
Availability of a visible poster with an operational flow-chart on the management of PPH In the delivery room	13 (54%)	11 (46%)	23 (96%)	1 (4%)
Availability of a kit (drugs, devices, etc.) dedicated to bleeding emergencies in the delivery room	19 (79%)	5 (21%)	24 (100%)	0
Organisation of multi-professional training on the management of obstetric emergencies, including PPH, involving gynaecologists, anaesthesiologists, midwives, and nurses simultaneously	19 (79%)	5 (21%)	22 (92%)	2 (8%)
Periodic multi-professional simulations of PPH treatment in place (missing = 2)	10 (42%)	12 (50%)	14 (58%)	8 (33%)
Adoption of a structured way to manage communication with family members in the event of a haemorrhagic emergency	5 (21%)	19 (79%)	11 (46%)	13 (54%)
Routinely offer to the woman and partner of a counselling on the events characterizing the bleeding, including the risk for future pregnancies before discharge (missing = 7)	5 (21%)	12 (50%)	10 (42%)	7 (29%)
Periodic clinical audits to assess cases of severe PPH (>1500 mL) (missing = 2)	7 (29%)	15 (63%)	15 (63%)	7 (29%)

**Table 2 ijerph-20-05297-t002:** Women’s socio-demographic and obstetric characteristics.

Variable	N = 1100	
	n	%
Age in years		
<35	591	53.8
35–39	370	33.6
≥40	139	12.6
Pre-pregnancy BMI		
<18.5	68	6.6
18.5–24.9	634	61.6
25.0–29.9	221	21.5
≥30	106	10.3
Missing	71	6.5
Citizenship		
Not Italian	215	21.1
Italian	803	78.9
Missing	82	7.5
Educational level		
<18.5	10	1.1
18.5–24.9	209	22.0
25.0–29.9	368	38.6
≥30	365	38.3
Missing	148	13.5
Parity		
Primipara	464	43.0
Multipara	615	57.0
Missing	21	1.9
Multiple pregnancy	77	7.0
Hysterotomy	213	19.4
Previous CS	160	14.6
Previous hysterotomy	109	9.9
Induction of labour	325	29.5
Oxytocin augmentation	314	28.5
Pharmacological analgesia	343	31.2
Mode of birth		
Vaginal birth	597	54.3
Instrumental vaginal birth	68	6.2
Elective CS	213	19.4
Urgent CS	186	16.9
Emergency CS	36	3.3

BMI: body mass index; CS: caesarean section.

**Table 3 ijerph-20-05297-t003:** Management of postpartum haemorrhage cases by blood loss volume.

Variable		Total (N = 1100)		1000–1500 mL (n = 833)		1501–2000 mL (n = 186)		>2000 mL (n = 81)	
		n	%	n	%	n	%	n	%
Use of graphic charts for monitoring vital signs postpartum	Yes	417	38.8	297	36.6	84	45.9	36	45.0
	No	657	61.2	514	63.4	99	54.1	44	55.0
Missing		26	2.4	22	2.6	3	1.6	1	0.0
Use of at least a recommended method for blood loss estimate	Yes	1082	98.8	824	99.2	179	97.8	79	97.5
	No	13	1.2	7	0.8	4	2.2	2	2.5
Missing		5	0.5	2	0.2	3	1.6	-	-
Availability of PT ratio during bleeding	Yes	557	50.9	388	46.7	119	64.3	50	63.3
	No	537	49.1	442	53.3	66	35.7	29	36.7
Missing		6	0.5	3	0.4	1	0.5	2	2.5
Availability of APTT ratio during bleeding	Yes	622	57.0	433	52.4	128	69.2	61	77.2
	No	468	43.0	393	47.6	57	30.8	18	22.8
Missing		10	0.9	7	0.8	1	0.5	2	2.5
Tranexamic acid	Yes	635	57.8	438	52.6	130	69.9	67	82.7
	No	463	42.1	393	47.2	56	30.1	14	17.3
Missing		2	0.2	2	0.2	-	-	-	-
Prostaglandins	Yes	778	70.9	645	77.6	97	52.2	36	44.4
	No	320	29.1	186	22.4	89	47.8	45	55.6
Missing		2	0.2	2	0.2	-	-	-	-
Balloon tamponade	Yes	165	15.6	83	10.4	49	27.2	33	43.4
	No	892	84.4	718	89.6	131	72.8	43	56.6
Missing		43	3.9	32	3.8	6	3.2	5	6.2
Surgical procedures and/or artery embolisation	Yes	104	9.5	50	6.2	23	12.7	31	39.2
	No	957	87.0	751	93.8	158	87.3	48	60.8
Missing		39	3.5	32	3.8	5	2.7	2	2.5
Transfusion of packed red blood cells	Yes	246	22.5	100	12.1	78	41.9	68	85.0
	No	848	77.5	728	87.9	108	58.1	12	15.0
Missing		6	0.5	5	0.6	-	-	1	0.0

## Data Availability

Data available on request due to restrictions.

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
