# Peer review of "Uptake and Adherence to National Guidelines on Postpartum Haemorrhage in Italy: The MOVIE before–after Observational Study"

_ijerph, 2023, doi:10.3390/ijerph20075297_

Round 1

Reviewer 1 Report

This is a clear, concise, and well-written manuscript. I would suggest to add a few sentences in the Introduction section about PPH risk factors, diagnosis and treatment.

Author Response

We thank the reviewer for the suggestion. We added the following sentences:

“Although there are several known risk factors for postpartum haemorrhage (PPH), including multiple pregnancy, prolonged third stage of labour, and retained placenta [5,6], many cases of PPH occur unexpectedly [5,6]. Early clinical and/or instrumental diagnosis are fundamental for prompt treatment. Mechanical measures to stimulate uterine contraction represent the first-line management of PPH [5,6]. Pharmacological (i.e. oxytocic drugs such as oxytocin, ergomentrine, and carboprost, or tranexamic acid) and surgical (e.g. balloon tamponade and compression sutures) measures are effective treatment that can be administered both in sequence or simultaneously [5,6].”

Reviewer 2 Report

Dear Authors, I congratulate you on your significant effort; I hope to see another report and adhesion afterward. It is so challenging to change attitudes. The reference 10 must be left justified

Best regards,

The reviewer

Author Response

We checked reference 10 and we noticed it was already left justified.

Reviewer 3 Report

The study entitled "Uptake and adherence to national guidelines on postpartum 2 haemorrhage in Italy, the MOVIE before-after observational 3 study" discusses the adherence to the PPM guidelines and clinical management of women with PPH.

Indeed it is well written and concise study which is designed carefully with novelty. The results are presented adequately, analyzed well and suitable for publication in the present form.

The study involved 6 Italian regions (between 2019-2020) to assess the adherence to the  national guideline on post-partum haemorrhage (PPH) and discuss the clinical management of haemorrhagic events in selected maternity units.

The study is novel and important because PPH is a major cause of maternal morbidity and mortality in both developed and developing countries. Also these kinds of studies will help to generate a comprehensive approach and different methodologies to minimize the risk of PPH.

Since this study focuses on a specific project "Monitoring and evaluation of the implementation of evidence-based recommendations for the management of obstetric haemorrhagic emergencies" and adherence to PPH guidelines before and after this project,  it will add new methodological approaches to improve theses kind of projects and to better deal with PPH.

The manuscript and study design seems good to me however, it would be good if authors include separate tables regarding the results section "3.4-3.6" to have a clear results interpretation.

Author Response

We thank the reviewer for their comment. We noticed the title of table 3 was wrong and we corrected it in “Management of postpartum haemorrhage cases by blood loss volume”. However, we find it difficult to split the table because of the little information contained in it.

Reviewer 4 Report

The manuscript is clear and scientifically sound. The topic is crucial for the field of perinatal care. The results are presented and interpreted appropriately. The conclusions are coherent. 

Two regions (Campania and Sicily) were excluded from the study due to < 50% coverage. Could the authors comment in the discussion, why they had problem to uptake and adhere to national guidelines on postpartum hemorrhage?

Author Response

We thank the reviewer for their comment, and we would like to clarify that we excluded these two Regions due to unsatisfying coverage. For this reason, we cannot speculate on these Regions’ guidelines uptake and adherence.